# Training Large Language Models to Reason in a Continuous Latent Space

**Shibo Hao**[1,2*], **Sainbayar Sukhbaatar**[1], **DiJia Su**[1], **Xian Li**[1], **Zhiting Hu**[2],
**Jason Weston**[1], **Yuandong Tian**[1]

[1]FAIR at Meta,   [2]UC San Diego

 https://github.com/facebookresearch/coconut

## Abstract

Large language models (LLMs) are restricted to reason in the "language space", where they typically express the reasoning process with a chain-of-thought (CoT) to solve a complex reasoning problem. However, we argue that language space may not always be optimal for reasoning. For example, some critical tokens require complex planning, making them difficult to compute in a single forward pass, while many other tokens contribute little to the actual reasoning process. To explore the potential of LLM reasoning in an unrestricted latent space instead of using natural language, we introduce a new paradigm COCONUT (Chain of Continuous Thought). We utilize the last hidden state of the LLM as a representation of the reasoning state (termed "continuous thought"). Rather than decoding this into a word token, we feed it back to the LLM as the subsequent input embedding directly in the continuous space. This latent reasoning paradigm leads to the emergence of an advanced reasoning pattern: the continuous thought can encode multiple alternative next reasoning steps, allowing the model to perform a breadth-first search (BFS) to solve the problem, rather than prematurely committing to a single deterministic path like CoT. COCONUT outperforms CoT on certain logical reasoning tasks that require substantial search during planning, and shows a better trade-off between accuracy and efficiency. We hope these findings demonstrate the promise of latent reasoning and offer insights for future research.

## 1 Introduction

Large language models (LLMs) have demonstrated remarkable reasoning abilities, emerging from extensive pretraining on human languages (Dubey et al., 2024; Achiam et al., 2023). While next token prediction is an effective training objective, it imposes a fundamental constraint on the LLM as a reasoning machine: the explicit reasoning process of LLMs must be generated in word tokens. For example, a prevalent approach, known as chain-of-thought (CoT) reasoning (Wei et al., 2022), involves prompting or training LLMs to generate solutions step-by-step using natural language. However, this stands in stark contrast to findings from studies on human cognition. Neuroimaging studies have consistently shown that the language network – a set of brain regions responsible for language comprehension and production – remains largely inactive during various reasoning tasks (Amalric & Dehaene, 2019; Monti et al., 2012; 2007; 2009; Fedorenko et al., 2011). Further evidence indicates that human language is optimized for communication rather than reasoning (Fedorenko et al., 2024).

A significant issue arises when LLMs use language for reasoning: the amount of reasoning required for each particular language token varies greatly, yet current LLM architectures allocate nearly the same computing budget for predicting every token. Some critical tokens require complex planning and pose huge challenges to LLMs, while most tokens in a

---

*Work done at Meta.

Figure 1: A comparison of Chain of Continuous Thought (COCONUT) with Chain-of-Thought (CoT). In CoT, the model generates the reasoning process as a word token sequence (e.g., $[x_i, x_{i+1}, ..., x_{i+j}]$ in the figure). COCONUT regards the last hidden state as a representation of the reasoning state (termed "continuous thought"), and directly uses it as the next input embedding. This allows the LLM to reason in an unrestricted latent space instead of a language space.

reasoning chain are generated solely for fluency, offering minimal contribution to the reasoning process and consequently reducing overall efficiency. While previous work has attempted to fix these problems by performing additional reasoning before generating some critical tokens (Zelikman et al., 2024) or encouraging LLMs to generate succinct reasoning chains (Madaan & Yazdanbakhsh, 2022; Nayab et al., 2024; Han et al., 2024), these solutions remain constrained within the language space and do not solve the fundamental problems. On the contrary, it would be ideal for LLMs to have the freedom to reason without any language constraints, and then translate their findings into language only when necessary.

In this work we instead explore LLM reasoning in a latent space by introducing a novel paradigm, COCONUT (Chain of Continuous Thought). It involves a simple modification to the traditional CoT process: instead of mapping between hidden states and language tokens using the language model head and embedding layer, COCONUT directly feeds the last hidden state (a continuous thought) as the subsequent input embedding for the next token (Figure 1). This modification frees the reasoning from being within the language space, and the system can be optimized end-to-end by gradient descent. To enhance the training of latent reasoning, we employ a multi-stage training strategy inspired by Deng et al. (2024), which effectively utilizes language reasoning chains to guide the training process.

This proposed method leads to an efficient reasoning pattern: Unlike language-based reasoning, continuous thoughts in COCONUT can encode multiple potential next steps simultaneously, allowing for a reasoning process akin to breadth-first search (BFS). While the model may not initially make the correct decision, it can maintain many possible options within the continuous thoughts and progressively eliminate incorrect paths through reasoning, guided by some implicit value functions. This advanced reasoning mechanism surpasses traditional CoT, even though the model is not explicitly trained or instructed to operate in this manner, as seen in previous works (Yao et al., 2023; Hao et al., 2023).

We further validate the feasibility of latent reasoning through additional analyses on three datasets. On other planning-intensive tasks, such as ProntoQA (Saparov & He, 2022) and our newly introduced ProsQA, COCONUT consistently achieves higher accuracy than CoT while generating fewer tokens. For math reasoning tasks on GSM8k (Cobbe et al., 2021), augmenting the LLM with six continuous thoughts doubles its performance. Additionally, COCONUT surpasses the strong baseline *iCoT* (Deng et al., 2024) and offers a superior trade-off between accuracy and efficiency compared to CoT. We believe these results underscore the significant potential of latent reasoning and offer valuable insights to guide future research.

## 2 Related Work

**Chain-of-thought (CoT) reasoning.** We use the term chain-of-thought broadly to refer to methods that generate an intermediate reasoning process in language before outputting the

final answer. This includes prompting LLMs (Wei et al., 2022; Khot et al., 2022; Zhou et al., 2022), or training LLMs to generate reasoning chains, either with supervised finetuning (Yue et al., 2023; Yu et al., 2023) or reinforcement learning (Wang et al., 2024; Havrilla et al., 2024; Shao et al., 2024; Yu et al., 2024a; Guo et al., 2025). Madaan & Yazdanbakhsh (2022); Nayab et al. (2024); Han et al. (2024) designed methods to make LLMs generate shorter reasoning chains. Recent theoretical analyses have demonstrated the usefulness of CoT from the perspective of model expressivity (Feng et al., 2023; Merrill & Sabharwal, 2023; Li et al., 2024). By employing CoT, the effective depth of the transformer increases because the generated outputs are looped back to the input (Feng et al., 2023). These analyses, combined with the established effectiveness of CoT, motivated our design that feeds the continuous thoughts back to the LLM as the next input embedding. While CoT has proven effective for certain tasks, its autoregressive generation nature makes it challenging to mimic human reasoning on more complex problems (LeCun, 2022; Hao et al., 2023), which require planning and search. There are works that equip LLMs with explicit tree search algorithms (Xie et al., 2023; Yao et al., 2023; Hao et al., 2024), or train the LLM on search dynamics and trajectories (Lehnert et al., 2024; Gandhi et al., 2024; Su et al., 2024). In our analysis, we find that after removing the constraint of a language space, a new reasoning pattern similar to BFS emerges, though the model is not explicitly trained this way.

**Latent reasoning in LLMs.** Previous works mostly define latent reasoning in LLMs as the hidden computation in transformers (Yang et al., 2024; Biran et al., 2024). Yang et al. (2024) constructed a dataset of two-hop reasoning problems and discovered that it is possible to recover the intermediate variable from the hidden representations. Biran et al. (2024) further proposed to intervene the latent reasoning by "back-patching" the hidden representation. Shalev et al. (2024) discovered parallel latent reasoning paths in LLMs. Another line of work has discovered that, even if the model generates a CoT to reason, the model may actually utilize a different latent reasoning process. This phenomenon is known as the unfaithfulness of CoT reasoning (Wang et al., 2022; Turpin et al., 2024). To enhance the latent reasoning of LLMs, previous research proposed to augment it with additional tokens. Goyal et al. (2023) pretrained the model by randomly inserting a learnable <pause> tokens to the training corpus. This improves LLM's performance on a variety of tasks, especially when followed by supervised finetuning with <pause> tokens. On the other hand, Pfau et al. (2024) further explored the usage of filler tokens, e.g., "...", and concluded that they work well for highly parallelizable problems. However, Pfau et al. (2024) mentioned these methods do not extend the expressivity of the LLM like CoT; hence, they may not scale to more general and complex reasoning problems. Wang et al. (2023) proposed to predict a planning token as a discrete latent variable before generating the next reasoning step. Recently, it has also been found that one can "internalize" the CoT reasoning into latent reasoning in the transformer with knowledge distillation (Deng et al., 2023) or a special training curriculum which gradually shortens CoT (Deng et al., 2024). Yu et al. (2024b) also proposed to distill a model that can reason latently from data generated with complex reasoning algorithms. These training methods can be combined to our framework, and specifically, we find that breaking down the learning of continuous thoughts into multiple stages, inspired by iCoT (Deng et al., 2024), is very beneficial for the training. Other work explores alternative architectures for latent reasoning, including looped transformers (Giannou et al., 2023; Fan et al., 2024), diffusion models in sentence embedding space (Barrault et al., 2024), depth-recurrent models (Geiping et al., 2025) or energy-based models (Gladstone et al., 2025). Different from these works, we focus on general multi-step reasoning tasks and aim at investigating the unique properties of latent reasoning in comparison to language space. Zhu et al. (2025) developed a theoretical construction of a 2-layer transformer with COCONUT, leveraging the idea that continuous thoughts serve as superpositional representations of multiple reasoning paths. It proves to solve the directed graph reachability problem more efficiently than the best-known theoretical results based on discrete CoT.

# 3    COCONUT: Chain of Continuous Thought

In this section, we introduce our new paradigm COCONUT (Chain of Continuous Thought) for reasoning in an unconstrained latent space. We begin by introducing the background and

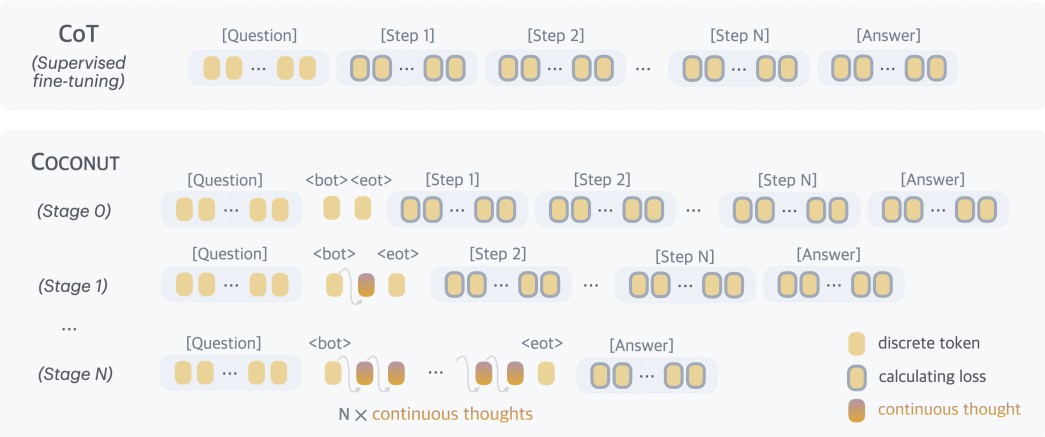

Figure 2: Training procedure of Chain of Continuous Thought (COCONUT). Given training data with language reasoning steps, at each training stage we integrate $c$ additional continuous thoughts ($c = 1$ in this example), and remove one language reasoning step. The cross-entropy loss is then used on the remaining tokens after continuous thoughts.

notation we use for language models. For an input sequence $x = (x_1, ..., x_T)$, the standard large language model $\mathcal{M}$ can be described as:

$$H_t = \text{Transformer}(E_t)$$
$$\mathcal{M}(x_{t+1} \mid x_{\leq t}) = \text{softmax}(Wh_t)$$

where $E_t = [e(x_1), e(x_2), ..., e(x_t)]$ is the sequence of token embeddings up to position $t$; $H_t \in \mathbb{R}^{t \times d}$ is the matrix of the last hidden states for all tokens up to position $t$; $h_t$ is the last hidden state of position $t$, i.e., $h_t = H_t[t, :]$; $e(\cdot)$ is the token embedding function; $W$ is the parameter of the language model head.

**Method Overview.** In the proposed COCONUT method, the LLM switches between the "language mode" and "latent mode" (Figure 1). In language mode, it operates as a standard language model, autoregressively generating the next token. In latent mode, it directly uses the last hidden state as the next input embedding. This last hidden state represents the current reasoning state, termed as a "continuous thought". Special tokens <bot> and <eot> are employed to mark the beginning and end of the latent thought mode.

**Training Procedure.** In this work, we consider a problem-solving scenario where the model receives a question as input and is expected to generate an answer through a reasoning process. We leverage language CoT data to supervise continuous thought by implementing a multi-stage training curriculum inspired by Deng et al. (2024). As shown in Figure 2, in the initial stage, the model is trained on regular CoT instances. Assume the total number of training stages (excluding the initial stage) is set to $N$ ($N = 6$ in this example). In the subsequent stages, at the $k$-th stage, the first $k$ reasoning steps in the CoT are replaced with $k \times c$ continuous thoughts[1], where $c$ is a hyperparameter controlling the number of latent thoughts replacing a single language reasoning step. Following Deng et al. (2024), we also reset the optimizer state when training stages switch.

During the training process, we optimize the negative log-likelihood loss as usual, but mask the loss on questions and latent thoughts. It is important to note that the objective does **not** encourage the continuous thought to *compress the removed language thought*, but rather to *facilitate the prediction of future reasoning*. Therefore, it's possible to learn more effective representations of reasoning steps compared to human language.

---

[1]If a language reasoning chain is shorter than $k$ steps, then all the language thoughts will be removed.

**Training Details.** Our proposed continuous thoughts are fully differentiable and allow for back-propagation. We perform $n + 1$ forward passes when $n$ latent thoughts are scheduled in the current training stage, computing a new latent thought with each pass and finally conducting an additional forward pass to obtain a loss on the remaining text sequence. While we have avoided redundant computations by reusing the KV cache across each forward pass, the sequential computation of COCONUT poses challenges on parallelism for the existing training infrastructure. Further optimizing the training efficiency of COCONUT remains an important direction for future research (Pöppel et al., 2025).

**Inference Process.** The inference process for COCONUT is analogous to standard language model decoding, except that in latent mode, we directly feed the last hidden state as the next input embedding. One practical challenge is determining when the model should switch between latent and language modes. As we focus on the problem-solving setting, we insert a `<bot>` token immediately following the question tokens. For `<eot>`, we consider two potential strategies: a) train a binary classifier on latent thoughts to enable the model to autonomously decide when to terminate the latent reasoning, or b) always pad the latent thoughts to a constant length. We found that both approaches work comparably well. Therefore, we use the second option in our experiment for simplicity, unless specified otherwise.

## 4 Continuous Space Enables Latent Tree Search

In this section, we provide a proof of concept on the advantage of continuous latent space reasoning. On ProsQA, a new dataset that requires extensive planning ability, COCONUT outperforms language space CoT reasoning. Interestingly, our analysis indicates that the continuous representation of reasoning can encode multiple alternative next reasoning steps. This allows the model to perform a breadth-first search (BFS) to solve the problem, instead of prematurely committing to a single deterministic path like language CoT.

We start by introducing the experimental setup (Section 4.1). By leveraging COCONUT's ability to switch between language and latent space reasoning, we are able to control the model to interpolate between fully latent reasoning and fully language reasoning and test their performance (Section 4.2). This also enables us to interpret the latent reasoning process as tree search (Section 4.3). Based on this perspective, we explain why latent reasoning can make the decision easier for LLMs (Section 4.4).

### 4.1 Experimental Setup

**Dataset.** We introduce ProsQA (Proof with Search Question-Answering), a new logical reasoning dataset. A visualized example is shown in Figure 4. Each instance in ProsQA consists of a directed acyclic graph (DAG) of logical relationships between concepts, presented as natural language statements. The task requires models to determine logical relationships by finding valid paths through this graph, demanding sophisticated planning and search strategies. Unlike previous logical reasoning datasets like ProntoQA (Saparov & He, 2022), ProsQA's DAG structure introduces complex exploration paths, making it particularly challenging for models to identify the correct reasoning chain. More comprehensive details about the dataset construction and characteristics can be found in Appendix A.

**Experimental Setup.** We use a pre-trained GPT-2 model as the base model for all experiments. The learning rate is set to $1 \times 10^{-4}$ while the effective batch size is 128. We train a COCONUT model following the training procedure in Section 3. Since the maximum reasoning steps in ProsQA is 6, we set the number of training stages to $N = 6$ in the training procedure. In each stage, we train the model for 5 epochs, and stay in the last stage until the 50 epochs. The checkpoint with the best accuracy in the last stage is used for evaluation. As reference, we report the performance of (1) *CoT*: the model is trained with CoT data, and during inference, the model will generate a complete reasoning chain to solve the problem. (2) *no-CoT*: the model is trained with only the question and answer pairs, without any reasoning steps. During inference, the model will output the final answer directly.

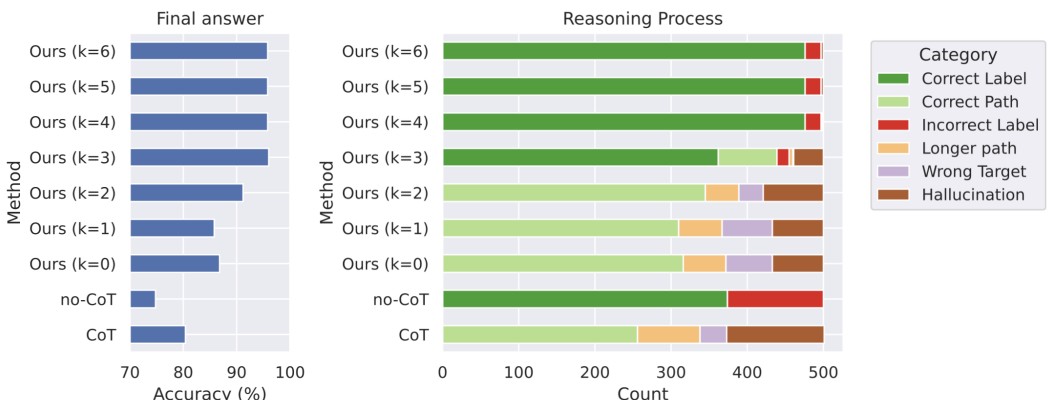

Figure 3: The accuracy of final answer (left) and reasoning process (right) of multiple variants of COCONUT and baselines on ProsQA.

To understand the properties of latent and language reasoning space, *we manipulate the model to switch between fully latent reasoning and fully language reasoning*, by manually setting the position of the <eot> token during inference. When we enforce COCONUT to use $k$ continuous thoughts, the model is expected to output the remaining reasoning chain in language, starting from the $k + 1$ step. In our experiments, we test variants of COCONUT on ProsQA with $k \in \{0, 1, 2, 3, 4, 5, 6\}$. Note that all these variants only differ in inference time while sharing the same model weights.

**Metrics.** We apply two sets of evaluation metrics. One of them is based on the correctness of the *final answer*, regardless of the reasoning process. It is also the main metric used in the later sections (Section 5.3). To enable fine-grained analysis on ProsQA, we define another metric on the *reasoning process*. We classify a reasoning chain into (1) **Correct Path**: The output is one of the shortest paths to the correct answer. (2) **Longer Path**: A valid path that correctly answers the question but is longer than the shortest path. (3) **Hallucination**: The path includes nonexistent edges or is disconnected. (4) **Wrong Target**: A valid path in the graph, but the destination node is not the one being asked. These four categories naturally apply to the output from COCONUT ($k = 0$) and *CoT*, which generate the full path. For COCONUT with $k > 0$ that outputs only partial paths in language (with the initial steps in continuous reasoning), we classify the reasoning as a Correct Path *if a valid explanation can complete it*. Also, we define Longer Path and Wrong Target for partial paths similarly. If no valid explanation completes the path, it's classified as Hallucination. In *no-CoT* and COCONUT with larger $k$, the model may only output the final answer without any partial path, and it falls into (5) **Correct Label** or (6) **Incorrect Label**. These six categories cover all cases without overlap.

## 4.2 Overall Results

Figure 3 presents a comparative analysis of various reasoning methods evaluated on ProsQA. The model trained using *CoT* frequently hallucinates non-existent edges or outputs paths leading to incorrect targets, resulting in lower answer accuracy. In contrast, COCONUT, which leverages continuous space reasoning, demonstrates improved accuracy as it utilizes an increasing number of continuous thoughts. Additionally, the rate of correct reasoning processes (indicated by "Correct Label" and "Correct Path") significantly increases. At the same time, there is a notable reduction in instances of "Hallucination" and "Wrong Target," issues that typically emerge when the model makes mistakes early in the reasoning process.

This improvement can be attributed to the inherent advantage of latent reasoning: it mitigates premature commitment to definitive choices by allowing the model to progressively refine its decisions through subsequent steps. Consequently, incorrect options are systematically eliminated over time, culminating in higher final accuracy. An intuitive demonstration of this benefit is provided by the case study depicted in Figure 4.

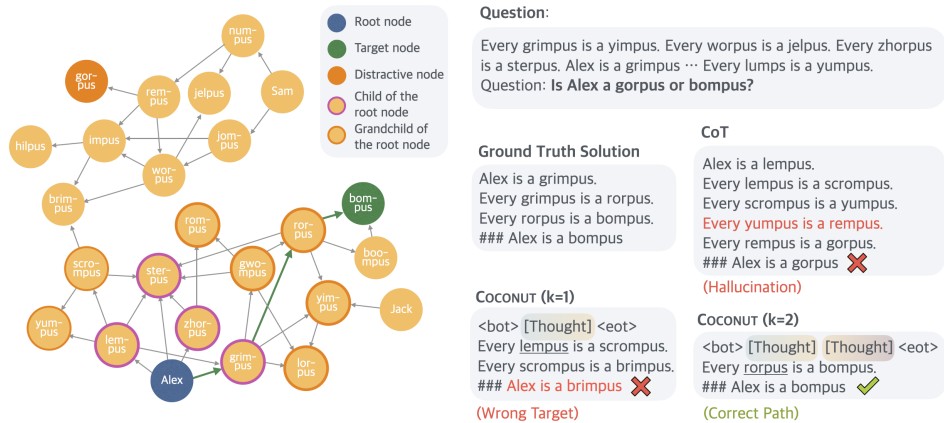

Figure 4: A case study of ProsQA. The model trained with *CoT* hallucinates an edge (*Every yumpus is a rempus*) after getting stuck in a dead end. COCONUT (k=1) outputs a path that ends with an irrelevant node. COCONUT (k=2) solves the problem correctly.

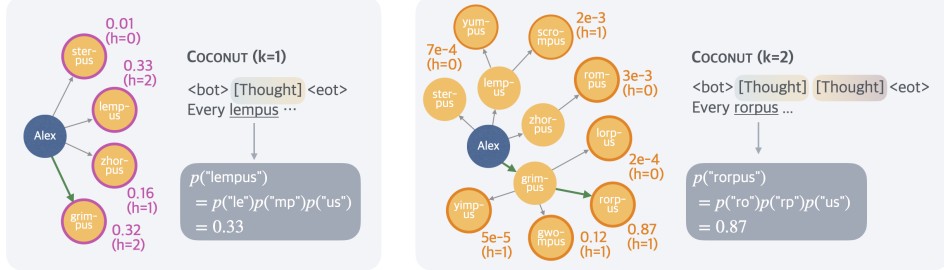

Figure 5: An illustration of the latent search trees. The example is the same test case as in Figure 4. We show the probability of the first concept predicted by the model following latent thoughts (e.g., "lempus" in the left figure). It is calculated as the multiplication of the probability of all tokens within the concept. The height of a node (denoted as $h$ in the figure) is defined as the shortest distance to any leaf node in the graph.

## 4.3 Interpreting the Latent Reasoning as Tree Search

To better understand COCONUT, we probe the latent reasoning process by forcing the model to explicitly generate language reasoning steps following intermediate continuous thoughts (Figure 5). Using the example presented in Figure 4, at the initial reasoning step, the model must select which immediate child node of 'Alex' to consider next, specifically from the set lempus, sterpus, zhorpus, grimpus. The distribution over these candidate next steps is visualized in Figure 5, left. In the subsequent reasoning step, these nodes expand further into an extended set of potential paths, including all grandchildren of 'Alex' (Figure 5, right).

We define the predicted probability of a concept following continuous thoughts as a value function (Figure 5), estimating each node's potential for reaching the correct target. Interestingly, the reasoning strategy employed by COCONUT is not greedy: while "lempus" initially has the highest value (0.33) at the first reasoning step (Figure 5, left), the model subsequently assigns the highest value (0.87) to "rorpus," a child of "grimpus," rather than following "lempus" (Figure 5, right). This characteristic resembles a breadth-first search (BFS) approach, contrasting sharply with the greedy decoding typical of traditional CoT methods. The inherent capability of continuous representations to encode multiple candidate paths enables the model to avoid making immediate deterministic decisions. Importantly, this tree search pattern is not limited to the illustrated example, but constitutes a fundamental mechanism underlying the consistent improvement observed with larger values of $k$ in COCONUT.

### 4.4 Why is a Latent Space Better for Planning?

Building upon the tree search perspective, we further examine why latent reasoning benefits planning tasks—specifically, why maintaining multiple candidate paths and postponing deterministic decisions enhances reasoning performance. Our hypothesis is that nodes explored in the early reasoning stages are inherently more challenging to evaluate accurately because they are farther from the final target nodes. In contrast, nodes positioned closer to potential targets, having fewer subsequent exploration possibilities, can be assessed accurately with higher confidence.

To systematically test this, we define the height of a node as its shortest distance to any leaf node and analyze the relationship between node height and the model's estimated value. Ideally, a correct node—one that can lead to the target node—should receive a high estimated value, whereas an incorrect node—one that cannot lead to the target node—should receive a low value.

Empirical results across the test set (Figure 6) support our hypothesis: nodes with lower heights consistently receive more accurate and definitive probability evaluations. Conversely, nodes with greater heights exhibit more ambiguous evaluations, reflecting increased uncertainty.

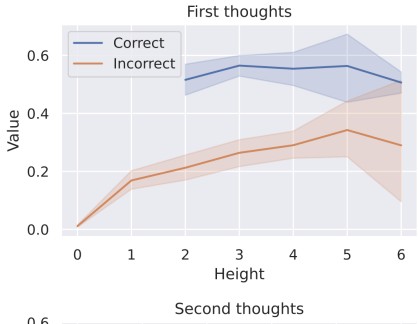
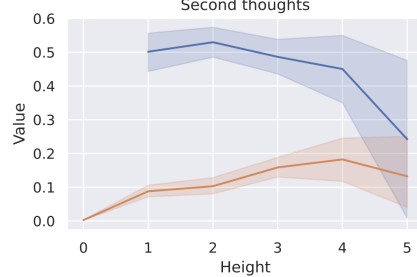

These findings underscore the advantage of latent space reasoning. By delaying deterministic decisions and allowing exploration to proceed toward terminal states, latent reasoning significantly enhances the model's ability to differentiate correct paths from incorrect ones, thereby improving performance on complex, planning-intensive tasks compared to traditional greedy methods. Motivated by these empirical observations, Zhu et al. (2025) presented a theoretical construction of a 2-layer transformer with COCONUT,

Figure 6: The correlation between the predicted value of correct/incorrect nodes and their heights.

demonstrating that it can solve the directed graph reachability problem using $O(n)$ steps, where $n$ is the number of vertices. This efficiency arises from the use of superposition to represent multiple explored vertices in parallel. On the contrary, the best known theoretical result for constant-depth transformers with discrete CoT requires $O(n^2)$ steps (Merrill & Sabharwal, 2023). Experiments also confirm that the construction aligns well with the empirical solution obtained via training dynamics.

## 5 Empirical Results with COCONUT

After showing the promising tree search pattern of COCONUT, we validate the feasibility of LLM reasoning in a continuous latent space through more comprehensive experiments, highlighting its better reasoning efficiency over language space, as well as its potential to enhance the model's expressivity.

### 5.1 Experimental Setup

**Math Reasoning.** We use GSM8k (Cobbe et al., 2021) as the dataset for math reasoning. It consists of grade school-level math problems. To train the model, we use a synthetic dataset generated by Deng et al. (2023). We use two continuous thoughts for each reasoning step (i.e., $c = 2$), and set the number of training stages to 3.

**Logical Reasoning.** Logical reasoning involves the proper application of known conditions to prove or disprove a conclusion using logical rules. We use the ProntoQA (Saparov & He, 2022) dataset, and our newly proposed ProsQA dataset, which is more challenging due to

| Method | GSM8k | | ProntoQA | | ProsQA | |
|---|---|---|---|---|---|---|
| | Acc. (%) | # Tokens | Acc. (%) | # Tokens | Acc. (%) | # Tokens |
| CoT | 42.9 ±0.2 | 25.0 | 98.8 ±0.8 | 92.5 | 77.5 ±1.9 | 49.4 |
| No-CoT | 16.5 ±0.5 | 2.2 | 93.8 ±0.7 | 3.0 | 76.7 ±1.0 | 8.2 |
| iCoT | 30.0* | 2.2 | 99.8 ±0.3 | 3.0 | 98.2 ±0.3 | 8.2 |
| Pause Token | 16.4 ±1.8 | 2.2 | 77.7 ±21.0 | 3.0 | 75.9 ±0.7 | 8.2 |
| COCONUT (Ours) | 34.1 ±1.5 | 8.2 | 99.8 ±0.2 | 9.0 | 97.0 ±0.3 | 14.2 |
| - w/o curriculum | 14.4 ±0.8 | 8.2 | 52.4 ±0.4 | 9.0 | 76.1 ±0.2 | 14.2 |
| - w/o thought | 21.6 ±0.5 | 2.3 | 99.9 ±0.1 | 3.0 | 95.5 ±1.1 | 8.2 |
| - pause as thought | 24.1 ±0.7 | 2.2 | 100.0 ±0.1 | 3.0 | 96.6 ±0.8 | 8.2 |

Table 1: Results on three datasets: GSM8k, ProntoQA and ProsQA. Higher accuracy indicates stronger reasoning ability, while generating fewer tokens indicates better efficiency. *The result is from Deng et al. (2024).

more distracting branches. We use one continuous thought for each reasoning step (i.e., $c = 1$), and set the number of training stages to 6.

More details of datasets and training settings are described in Appendix A and Appendix B.3.

## 5.2 Baselines and Variants of COCONUT

We consider the following baselines: (1) *CoT*, and (2) *No-CoT*, which were introduced in Section 4. (3) *iCoT* (Deng et al., 2024): The model is trained with language reasoning chains and follows a carefully designed schedule that "internalizes" CoT. As the training goes on, tokens at the beginning of the reasoning chain are gradually removed until only the answer remains. During inference, the model directly predicts the answer. (4) *Pause token* (Goyal et al., 2023): The model is trained using only the question and answer, without a reasoning chain. However, different from *No-CoT*, special <pause> tokens are inserted between the question and answer, which provides the model with additional computational capacity to derive the answer. The number of <pause> tokens is set the same as continuous thoughts in COCONUT.

We also evaluate some variants of COCONUT: (1) *w/o curriculum*, which directly trains the model in the last stage. The model uses continuous thoughts to solve the whole problem. (2) *w/o thought*: We keep the multi-stage training, but don't add any continuous latent thoughts. While this is similar to *iCoT* in the high-level idea, the exact training schedule is set to be consistent with COCONUT, instead of *iCoT*, for a strict comparison. (3) *Pause as thought*: We use special <pause> tokens to replace the continuous thoughts, and apply the same multi-stage training curriculum as COCONUT.

## 5.3 Results and Discussion

We show the overall results on all datasets in Table 1. Using continuous thoughts effectively enhances LLM reasoning over the No-CoT baseline. For example, by using 6 continuous thoughts, COCONUT achieves 34.1% accuracy on GSM8k, which significantly outperforms *No-CoT* (16.5%). We list several key conclusions from the experiments as follows. More discussions are in Appendix B.7.

**"Chaining" continuous thoughts enhances reasoning.** Language CoT proves to increase the effective depth of LLMs and enhance their expressiveness (Feng et al., 2023). Thus, generating more tokens serves as a way to inference-time scaling for reasoning (Guo et al., 2025; Snell et al., 2024). This desirable property holds naturally for COCONUT too. On GSM8k, COCONUT outperformed other architectures trained with similar strategies, including COCONUT (*pause as thought*) and COCONUT (*w/o thought*). Particularly, it surpasses the

latest baseline *iCoT* (Deng et al., 2024), which requires a more carefully designed training schedule.

Additionally, we experimented with adjusting the hyperparameter $c$, which controls the number of latent thoughts corresponding to one language reasoning step (Figure 7, II). As we increased $c$ from 0 to 1 to 2, the model's performance steadily improved.[2] This further validates the potential of continuous thoughts to scale up to harder problem. In two other synthetic tasks, we found that the variants of COCONUT (*w/o thoughts* or *pause as thought*), and the *iCoT* baseline also achieves impressive accuracy. This indicates that the model's computational capacity may not be the bottleneck in these tasks. In contrast, GSM8k involves more complex contextual understanding and modeling, placing higher demands on computational capability.

**Continuous thoughts are efficient representations of reasoning.** Compared to traditional CoT, CO-CONUT generates fewer tokens while achieving higher accuracy on ProntoQA and ProsQA (Table 1). Although COCONUT does not surpass *CoT* on GSM8k, it offers a superior trade-off between reasoning efficiency and accuracy (Figure 7, I). To illustrate this, we train a series of CoT models that progressively skip (or "internalize" (Deng et al., 2024)) the initial $m = \{0, 1, 2, 3, \text{ALL}\}$ reasoning steps, and plot their accuracy versus the number of generated tokens (labeled as "language" in the figure). These CoT models quickly lose accuracy as the token generation budget decreases. In contrast, by applying CO-CONUT training strategy—replacing each language reasoning step with two continuous thoughts—the accuracy drop is substantially mitigated, maintaining higher performance even when fewer tokens are generated. Another interesting observation is that, when we decode the first continuous thought, it often corresponds to possible intermediate variables in the calculation (Figure 9). This also suggests that the continuous thoughts are more efficient representations of reasoning.

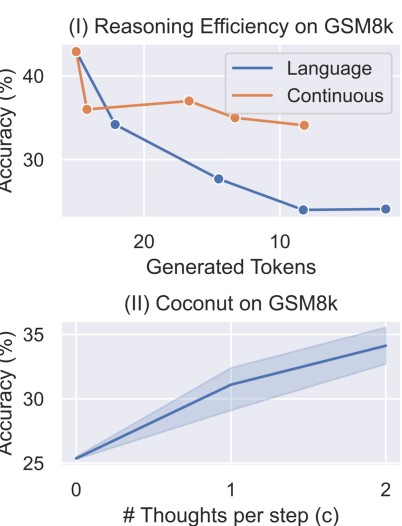

Figure 7: Efficiency comparison of reasoning space and COCONUT with difference $c$.

## 6    Conclusion

In this paper, we introduce COCONUT, a new paradigm for reasoning in continuous latent space. Experiments demonstrate that COCONUT effectively enhances LLM performance across a variety of reasoning tasks. Reasoning in latent space gives rise to advanced emergent behaviors, where continuous thoughts can represent multiple alternative next steps. This enables the model to perform BFS over possible reasoning paths, rather than prematurely committing to a single deterministic trajectory as in Chain-of-Thought (CoT) reasoning. Further research is needed to refine and scale latent reasoning to pretraining, which could improve generalization across a broader range of reasoning challenges. We hope our findings will spark continued exploration into latent reasoning, ultimately advancing the development of more capable machine reasoning systems.

## Acknowledgements

The authors express their sincere gratitude to Jihoon Tack for his valuable discussions throughout the course of this work.

---

[2]We discuss the case of larger $c$ in Appendix B.8.

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

# A    Datasets

## A.1    Examples

We provide some examples of the questions and CoT solutions for the datasets used in our experiments.

---

**GSM8k**

```
Question = "John cuts his grass to 2 inches. It grows .5 inches per month.
When it gets to 4 inches he cuts it back down to 2 inches. It cost $100 to
get his grass cut. How much does he pay per year?"
Steps = ["«4-2=2»", "«2/.5=4»", "«12/4=3»", "«100*3=300»"]
Answer = "300"
```

---

**ProntoQA**

```
Question = "Brimpuses are not luminous. Shumpuses are amenable. Each yumpus
is a lorpus. Gorpuses are shumpuses. Each zumpus is a grimpus. Gorpuses
are rompuses. Dumpuses are not floral. Lempuses are cold. Brimpuses are
impuses. Every lorpus is floral. Every rompus is transparent. Grimpuses
are muffled. Rompuses are yumpuses. Rompuses are wumpuses. Zumpuses are
fast. Wumpuses are bitter. Every sterpus is orange. Each lorpus is a vumpus.
Yumpuses are feisty. Each yumpus is a lempus. Gorpuses are snowy. Zumpuses
are gorpuses. Every lorpus is a sterpus. Stella is a brimpus. Stella is a
zumpus. True or false: Stella is not floral."
Steps = ["Stella is a zumpus. Zumpuses are gorpuses.", "Stella is a gorpus.
Gorpuses are rompuses.", "Stella is a rompus.  Rompuses are yumpuses.",
"Stella is a yumpus. Each yumpus is a lorpus.", "Stella is a lorpus. Every
lorpus is floral.", "Stella is floral."]
Answer = "False"
```

| # Nodes | # Edges | Len. of Shortest Path | # Shortest Paths |
|---------|---------|-----------------------|------------------|
| 23.0    | 36.0    | 3.8                   | 1.6              |

Table 2: Statistics of the graph structure in ProsQA.

| Dataset  | Training | Validation | Test |
|----------|----------|------------|------|
| GSM8k    | 385,620  | 500        | 1319 |
| ProntoQA | 9,000    | 200        | 800  |
| ProsQA   | 17,886   | 300        | 500  |

Table 3: Statistics of the datasets.

---

**ProsQA**

```
Question = "Every shumpus is a rempus. Every shumpus is a yimpus. Every
terpus is a fompus. Every terpus is a gerpus. Every gerpus is a brimpus.
Alex is a rempus. Every rorpus is a scrompus. Every rorpus is a yimpus.
Every terpus is a brimpus. Every brimpus is a lempus. Tom is a terpus.
Every shumpus is a timpus. Every yimpus is a boompus. Davis is a shumpus.
Every gerpus is a lorpus. Davis is a fompus. Every shumpus is a boompus.
Every shumpus is a rorpus. Every terpus is a lorpus. Every boompus is a
timpus. Every fompus is a yerpus. Tom is a dumpus. Every rempus is a rorpus.
Is Tom a lempus or scrompus?"
Steps = ["Tom is a terpus.", "Every terpus is a brimpus.", "Every brimpus
is a lempus."]
Answer = "Tom is a lempus."
```

---

### A.2 Construction of ProsQA

To construct the dataset, we first compile a set of typical entity names, such as "Alex" and "Jack," along with fictional concept names like "lorpus" and "rorpus," following the setting of ProntoQA (Saparov & He, 2022). Each problem is structured as a binary question: "Is [Entity] a [Concept A] or [Concept B]?" Assuming [Concept A] is the correct answer, we build a directed acyclic graph (DAG) where each node represents an entity or a concept. The graph is constructed such that a path exists from [Entity] to [Concept A] but not to [Concept B].

Algorithm 1 describes the graph construction process. The DAG is incrementally built by adding nodes and randomly connecting them with edges. To preserve the validity of the binary choice, with some probability, we enforce that the new node cannot simultaneously serve as a descendant to both node 0 and 1. This separation maintains distinct families of nodes and balances their sizes to prevent model shortcuts.

After the graph is constructed, nodes without parents are assigned entity names, while other nodes receive concept names. To formulate a question of the form "Is [Entity] a [Concept A] or [Concept B]?"", we designate node 0 in the graph as [Entity], a leaf node labeled 1 as [Concept A], and a leaf node labeled 2 as [Concept B]. This setup ensures a path from [Entity] to [Concept A] without any connection to [Concept B], introducing a moderately complex reasoning path. Finally, to avoid positional biases, [Concept A] and [Concept B] are randomly permuted in each question.

### A.3 Statistics

We show the size of all datasets in Table 3.

---

**Algorithm 1** Graph Construction for ProsQA

---

$edges \leftarrow \{\}$
$nodes \leftarrow \{0, 1\}$
$labels \leftarrow \{0 : 1, 1 : 2\}$
        ▷ Labels: 1 (descendant of node 0), 2 (descendant of node 1), 3 (both), 0 (neither).
$groups \leftarrow \{0 : \{\}, 1 : \{0\}, 2 : \{1\}, 3 : \{\}\}$
$idx \leftarrow 2$
**while** $idx < N$ **do**
                ▷ For each new node, randomly add edges from existing nodes
    $n\_in\_nodes \leftarrow \text{poisson}(1.5)$
    $rand \leftarrow \text{random}()$
    **if** $rand \leq 0.35$ **then**
        $candidates \leftarrow groups[0] \cup groups[1]$         ▷ Cannot be a descendant of node 1.
    **else if** $rand \leq 0.7$ **then**
        $candidates \leftarrow groups[0] \cup groups[2]$         ▷ Cannot be a descendant of node 0.
    **else**
        $candidates \leftarrow nodes$
    **end if**
    $n\_in\_nodes \leftarrow \min(\text{len}(candidates), n\_in\_nodes)$
    $weights \leftarrow [\text{depth\_to\_root}(c) \cdot 1.5 + 1 \; \forall c \in candidates]$
                ▷ Define sampling weights to prioritize deeper nodes.
            ▷ This way, the solution reasoning chain is expected to be longer.
    $in\_nodes \leftarrow \text{random\_choice}(candidates, n\_in\_nodes, \text{prob} = weights/\text{sum}(weights))$
    $cur\_label \leftarrow 0$
    **for** $in\_idx \in in\_nodes$ **do**
        $cur\_label \leftarrow cur\_label \;|\; labels[in\_idx]$         ▷ Update label using bitwise OR.
        $edges.\text{append}((in\_idx, idx))$
    **end for**
    $groups[cur\_label].\text{append}(idx)$
    $labels[idx] \leftarrow cur\_label$
    $nodes \leftarrow nodes \cup \{idx\}$
    $idx \leftarrow idx + 1$
**end while**

---

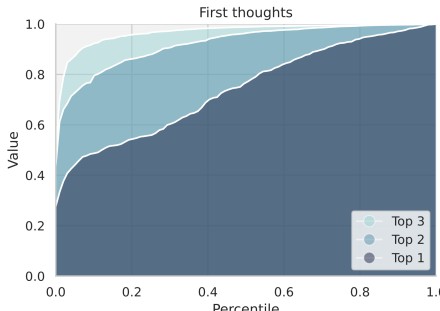 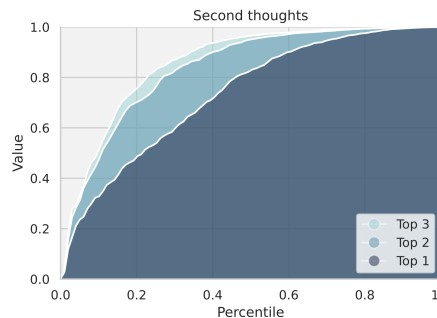

Figure 8: Analysis of parallelism in latent tree search. The left plot depicts the cumulative value of the top-1, top-2, and top-3 candidate nodes for the first thoughts, calculated across test cases and ranked by percentile. The significant gaps between the lines reflect the model's ability to explore alternative latent thoughts in parallel. The right plot shows the corresponding analysis for the second thoughts, where the gaps between lines are narrower, indicating reduced parallelism and increased certainty in reasoning as the search tree develops. This shift highlights the model's transition toward more focused exploration in later stages.

## B  More Discussion on Empirical Results

### B.1  COCONUT Generates Better Reasoning Chains

As shown in Figure 3, even when COCONUT is forced to generate a complete reasoning chain, the accuracy of the answers is still higher than *CoT*. The generated reasoning paths are also more accurate with less hallucination. From this, we can infer that the training method of mixing different stages improves the model's ability to plan ahead. The training objective of *CoT* always concentrates on the generation of the immediate next step, making the model "shortsighted". In later stages of COCONUT training, the first few steps are hidden, allowing the model to focus more on future steps. This is related to the findings of Gloeckle et al. (2024), where they propose multi-token prediction as a new pretraining objective to improve the LLM's ability to plan ahead.

### B.2  Analysis of Parallelism in Latent Tree Search

We present an analysis of the model's ability to explore alternative latent thoughts in parallel. As shown in Figure 8, the model makes use of the latent space to explore multiple paths in parallel, and this ability is more pronounced in the early stages of the search.

### B.3  Training Details

**Math Reasoning.**  By default, we use 2 latent thoughts (i.e., $c = 2$) for each reasoning step. We analyze the correlation between performance and $c$ in Section 5.3. The model goes through 3 stages besides the initial stage. Then, we have an additional stage, where we still use $3 \times c$ continuous thoughts as in the penultimate stage, but remove all the remaining language reasoning chain. This handles the long-tail distribution of reasoning chains longer than 3 steps. We train the model for 6 epochs in the initial stage, and 3 epochs in each remaining stage.

**Logical Reasoning.**  We use one continuous thought for every reasoning step (i.e., $c = 1$). The model goes through 6 training stages in addition to the initial stage, because the maximum number of reasoning steps is 6 in these two datasets. The model then fully reasons with continuous thoughts to solve the problems in the last stage. We train the model for 5 epochs per stage.

For all datasets, after the standard schedule, the model stays in the final training stage, until the 50th epoch. We select the checkpoint based on the accuracy on the validation set. For

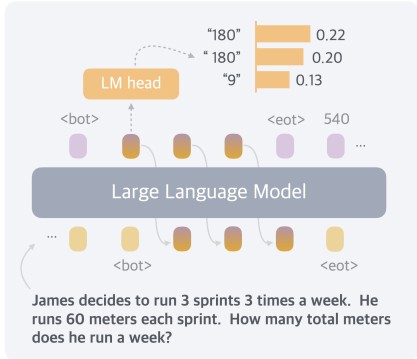

Figure 9: A case study where we decode the continuous thought into language tokens.

inference, we manually set the number of continuous thoughts to be consistent with their final training stage. We use greedy decoding for all experiments.

### B.4 Hyperparameter Searching

We perform a hyperparameter search over learning rates in the set $1 \times 10^{-3}, 1 \times 10^{-4}, 1 \times 10^{-5}$ and batch sizes in $32, 64, 128, 256$. The maximum number of training epochs is set to 50, as we observe convergence across all runs within this limit. Additionally, we tune the number of training epochs per stage for COCONUT over the set $1, 3, 5$ individually for each task.

### B.5 Clock-Time Reasoning Efficiency Metric

We present a clock-time comparison to evaluate reasoning efficiency. The reported values represent the average inference time per test case (in seconds), with a batch size of 1, measured on an Nvidia A100 GPU. For the no-CoT and CoT baselines, we employ the standard generate method from the `transformers`[3] library. Our results show that clock time is generally proportional to the number of newly generated tokens, as detailed in Figure 7.

| Method | GSM8k | ProntoQA | ProsQA |
|---|---|---|---|
| No-CoT | 0.03 | 0.03 | 0.08 |
| CoT | 0.26 | 0.85 | 0.47 |
| COCONUT | 0.09 | 0.11 | 0.15 |

Table 4: Inference time (in seconds) comparison across tasks and methods.

### B.6 Interpretation of Continuous Thoughts

In Figure 9, we show a case study where we decode the continuous thought into language tokens. The first continuous thought can be decoded into tokens like "180", " 180" (with a space), and "9". Note that, the reasoning trace for this problem should be $3 \times 3 \times 60 = 9 \times 60 = 540$, or $3 \times 3 \times 60 = 3 \times 180 = 540$. The interpretations of the first thought happen to be the first intermediate variables in the calculation. Moreover, it encodes a distribution of different traces into the continuous thoughts. This is consistent to the analysis in Section 4.3.

### B.7 More Discussions on Empirical Results

**Performance Differences among Different Datasets.** We discuss the performance differences among different datasets, to understand which tasks benefit more from latent reasoning.

---

[3] https://github.com/huggingface/transformers

- Real-World vs. Synthetic Domains: GSM8k represents a real-world, open-domain question-answering task. Unlike the synthetic datasets used in our study, it demands more complex contextual understanding and modeling, which can place greater demands on computational capabilities. This hypothesis is supported by the observation that COCONUT outperforms all other latent reasoning methods, and its accuracy steadily improves as the number of thoughts per step ($c$) increases from 0 to 2. Additionally, GSM8k requires diverse commonsense and world knowledge. This may give CoT an advantage, as it aligns closely with the pretraining objectives of the underlying language model, enabling it to better leverage its knowledge compared to COCONUT.

- Planning Requirements: Complex reasoning tasks often require the model to "look ahead" to determine whether a particular step is optimal (also known as planning). Among the datasets in our experiments, GSM8k involves grade-school-level math word problems that allow for intuitive judgment of the next reasoning step. Similarly, ProntoQA includes distracting branches of limited size, making it relatively straightforward to identify the correct next step. In contrast, ProsQA, based on a randomly generated Directed Acyclic Graph (DAG) structure, presents a significant challenge to the model's planning abilities. Our experimental results suggest that tasks requiring extensive planning benefit more from latent space reasoning (including COCONUT, some of its variants, and *iCoT*) than from reasoning using language tokens (*CoT*).

**The LLM still needs guidance to learn continuous thoughts**. In the ideal case, the model should learn the most effective continuous thoughts automatically through gradient descent on questions and answers (i.e., COCONUT *w/o curriculum*). However, from the experimental results, we found the models trained this way do not perform any better than *no-CoT*. With the multi-stage curriculum which decomposes the training into easier objectives, COCONUT is able to achieve top performance across various tasks. The multi-stage training also integrates well with pause tokens (COCONUT- *pause as thought*). Despite using the same architecture and similar multistage training objectives, we observed a small gap between the performance of *iCoT* and COCONUT *(w/o thoughts)*. The finer-grained removal schedule (token by token) and a few other tricks in *iCoT* may ease the training process. We leave combining *iCoT* and COCONUT as a future work. While the multi-stage training used for COCONUT has proven effective further research is definitely needed to develop better and more general strategies for learning reasoning in latent space, especially without the supervision from language reasoning chains

## B.8 Using More Continuous Thoughts

In Figure 7 (II), we present the performance of COCONUT on GSM8k using $c \in \{0, 1, 2\}$. When experimenting with $c = 3$, we observe a slight performance drop accompanied by increased variance. Analysis of the training logs indicates that adding three continuous thoughts at once – particularly during the final stage transition – leads to a sharp spike in training loss, causing instability. Future work will explore finer-grained schedules, such as incrementally adding continuous thoughts one at a time while removing fewer language tokens, as in iCoT (Deng et al., 2024). Additionally, combining language and latent reasoning—e.g., generating the reasoning skeleton in language and completing the reasoning process in latent space—could provide a promising direction for improving performance and stability.

## B.9 COCONUT with Larger Language Models

We experimented with COCONUT on GSM8k using Llama 3.2-3B and Llama 3-8B (Dubey et al., 2024) with $c = 1$. We train them for 3 epochs in Stage 0, followed by 1 epoch per subsequent stage. The results are shown in Table 5.

We observe consistent performance gains across both Llama 3.2-3B and Llama 3-8B models compared to the no-CoT baseline, though these improvements are not as pronounced as those previously demonstrated with GPT-2. One possible reason is that larger models have

| Model | no-CoT | COCONUT (Ours) |
|-------|--------|----------------|
| Llama 3.2-3B | 26.0 | 31.7 |
| Llama 3-8B | 42.2 | 43.6 |

Table 5: Experimental results of applying COCONUT to larger Llama models. We report performance comparisons between models without CoT reasoning (no-CoT) and our proposed COCONUT method.

already undergone extensive language-focused pre-training, making the transition to latent reasoning more challenging.

We emphasize that the primary goal of this paper is to highlight the promising attributes of latent-space reasoning and to initiate exploration in this new direction. Universally surpassing language-based CoT likely requires significant research efforts dedicated to **latent space pre-training**. We are encouraged by recent progress in this area (Geiping et al., 2025; Barrault et al., 2024). While these recent models provide scalable methods for latent representation learning, their latent spaces have not yet been explicitly optimized for reasoning. Integrating these recent advancements with COCONUT presents an exciting and promising avenue for future research.

