# OpenReview forum: "Training Large Language Models to Reason in a Continuous Latent Space"
_colmweb.org/COLM/2025/Conference — COLM 2025_

### Official Review · Reviewer_Hbwo · 2025-05-08

**Rating:** 7
**Confidence:** 3
**Ethics Flag:** 1

**Summary:**

The paper proposes a new approach for Continuous CoT, feeding the output token before the unembedding layer to the first layer at the next timestep, thereby creating a differentiable pathway across timesteps. The method leverages an iterative training procedure, where standard CoT steps are progressively replaced from the left with continuous ones. The authors analyze the behavior of the learned system, and draw connections with search-based procedures, showing the model can encode a distribution over next steps, rather than committing to a single one. This demonstration is aided by a reasoning dataset proposed by the authors, ProsQA.

The authors show that latent space reasoning is able to make a better use of the compute allocation across tokens, showing that for a fixed token budget, it outperforms standard CoT.

**Questions To Authors:**

1. Is the observed BFS-like behavior a property of Coconut on the given ProsQA dataset, or do you believe this behavior to emerge on other reasoning datasets not explicitly built around search ?
2. For figure 3, was greedy decoding to generate the potential valid generation ?

**Reasons To Accept:**

1. The authors explore the relatively new field of continuous reasoning, and show promising results.
2. The paper is clearly presented, and overall easy to follow. The authors do a good job at investigating the underlying mechanisms of Coconut, designing targeted experiments and ablations
3. The authors propose a new synthetic dataset, ProsQA, to aid understsanding of reasoning in search-based settings

**Reasons To Reject:**

1. The proposed training procedure is convoluted, requiring multiple training phases
2.  The connection to BFS is somewhat flaky; the authors argue that Coconut "allows the model to perform a breadth-first search (BFS) to solve the problem, instead of prematurely committing to a single deterministic path like language CoT." To this end, the authors show one example of a BFS-like behavior with the model. The authors allude to observing this beyond this single example. Is there a way to quantify this, or to show this repeating beyond a single example ? The current paper lacks strong evidence to support the BFS claim in its current form.

---

> ### Author Response · Authors · 2025-06-02
>
> Thank you for recognizing our clear presentation, targeted analyses, and the gains delivered by continuous latent reasoning on ProsQA. Below we address each of your questions.
>
> > Convoluted training procedure
>
> Our primary goal in this paper is to expose the benefits of latent reasoning—parallel search and higher reasoning efficiency. We would also like to mention that the current three‑stage curriculum is already simpler than some of the earlier work. For example, iCoT [1] removes CoT tokens one‑by‑one, leading to far longer training, and requires additional tricks like “removal smoothing,”. We expect future pre‑training in latent space to mitigate the need for dataset‑specific multi-stage training altogether.
>
> > Quantifying BFS
>
> Fig. 3 reports aggregate accuracy on the full ProsQA test set as we increase the number of continuous thoughts $(k → k + 1)$. The upward trend implies that, for many instances, adding one latent step flips an incorrect outcome to a correct one. For adjacent pair $(k, k + 1)$, especially when $k=1, 2$, there is a subset of testset items where correctness flips. By definition, every flipped case must follow the same logic:
>
> - With $k$ latent steps the model switches to language too early, commits to the “currently most promising” node, and later fails.
> - With $k + 1$ latent steps it keeps several branches alive for one more step and then follows the correct path.
>
> Figs. 4–5 present a randomly chosen instance from that subset to visualise this mechanism. The behaviour shown there recurs across all flipped cases, so the example is illustrative—not exceptional.
>
> Furthermore, Figure 6 quantifies the link between a node’s predicted value and its height (distance to a leaf), clarifying why extra continuous thoughts improve BFS performance. Figure 8 plots the exploration distribution of continuous thoughts and shows uncertainty shrinking with each search step. Both analyses use the entire ProsQA test set.
>
> > BFS-like behavior on other datasets
>
> We believe that many multi‑step reasoning tasks can be framed as path‑finding on a graph, and that Coconut naturally supports parallel exploration of such paths. Recent theoretical work [3, 4] also shows that the continuous thought architecture inherently enables this search behaviour.
>
> In our experiment, we chose ProsQA for analysis because its explicit DAG allows us to measure search properties such as height and frontier distribution without the confounds of natural‑language understanding. For open-domain tasks like GSM8K, we still see evidence that continuous thoughts encode multiple candidate paths (Fig. 9), but a clean search‑tree metric would be harder to define.
>
> > Greedy decoding
>
> We apply greedy decoding for all experiments in Figure 3. We use this setting because previous work [2] shows it typically performs well when using low temperature or greedy decoding for reasoning tasks.
>
> Thanks again for your valuable feedback!
>
> ## Reference
>
> [1] Deng et al., 2024, “From explicit cot to implicit cot: Learning to internalize cot step by step”
>
> [2] Wei et al., 2022, “Chain-of-Thought Prompting Elicits Reasoning in Large Language Models”
>
> [3] Gozeten et al., 2025, “Continuous Chain of Thought Enables Parallel Exploration and Reasoning”
>
> [4] Zhu et al., 2025, “Reasoning by Superposition: A Theoretical Perspective on Chain of Continuous Thought”

---

### Official Review · Reviewer_NeU4 · 2025-05-12

**Rating:** 6
**Confidence:** 4
**Ethics Flag:** 1

**Summary:**

The paper proposes a new reasoning paradigm for language models, called COCONUT (Chain of Continuous Thought), which allows reasoning in a continuous latent space rather than traditional language-based chains of thought (CoT). By using the last hidden state as an input embedding for subsequent reasoning steps, COCONUT avoids language constraints, enhancing reasoning efficiency. The authors use a multi-stage curriculum training scheme that gradually shifts from language-based reasoning to continuous thoughts, helping models learn effective latent reasoning representations. Experiment results with pre-trained GPT2 on synthetical datasets show that COCONUT is most effective in planning intensive tasks.

**Questions To Authors:**

N/A

**Reasons To Accept:**

1. One of the early works in latent CoT reasoning. Many follow-ups after this.

2. Sufficient analysis on accuracy, efficiency, and possible tree-search related explanation.

3. Show better performance on GSM8K than previous latent CoT methods.

**Reasons To Reject:**

1. Limited effectiveness in terms of performance on real-world reasoning task, GSM8K.

2. Only GPT2 model is used in the experiments.

3. The proposed method directly feeds the last layer output representation as the input embeddings, which makes the training procedure non-parallelizable sequence-wise, as in normal language models. Might affect scalability.

---

> ### Author Response · Authors · 2025-06-02
>
> Thank you for the thoughtful review and for highlighting our method’s potential impact, thorough analysis, and better empirical results. Below we address each of your questions.
>
> > More general reasoning tasks
> Regarding GSM8K performance, please see our general response for additional discussion. We agree that fully unlocking the potential of latent reasoning will require further research, and we are actively pursuing larger-scale latent space pretraining to close the gap.
>
> > Only GPT‑2 is used
>
> We experimented with COCONUT using Llama 3.2-3B and Llama 3-8B with $c=1$, training for 3 epochs for Llama 3.2-3B and 2 epochs in Stage 0, followed by 1 epoch per subsequent stage. The results are as follows:
>
> | Model | No‑CoT | Coconut |
> | - | - | - |
> | Llama‑3.2 3B | 26.0 | 31.7 |
> | Llama‑3  8B | 42.2 | 43.6 |
>
> Both backbones show clear gains over the no‑CoT baseline, though the boost is smaller than with GPT‑2. One hypothesis is that larger LLMs encode more knowledge for math reasoning problems learned from language-space pre-training, and therefore latent space pre-training might be crucial for coconut to achieve comparable or better performance to language space reasoning.
>
> > Scalability
>
> Though the current training method is sequential, we are looking into two potential paths to address this:
> - System‑level optimization: For example, FlashRNN [1] shows that careful kernel design (Triton/CUDA, register‑level tuning) can slash the overhead of recurrent computation.
> - Post‑training application. Recent work on latent space pretraining [2] applies a fixed sentence embedding, which enables parallel training at sequence level. Coconut could play the role of post‑training stage after such large-scale pre-training, in order to optimize the latent space for reasoning. This mirrors recent progress in online RL for LLMs—despite sequential rollouts, methods like OpenAI’s O1 [3] and DeepSeek’s R1 [4] deliver meaningful gains on long‑horizon reasoning.
>
> Thanks again for your valuable feedback!
>
> ## References
>
> [1] Pöppel et al., 2025, FlashRNN: I/O‑Aware Optimisation of Traditional RNNs on Modern Hardware
>
> [2] LCM Team, 2024, Large Concept Models: Language Modelling in a Sentence Representation Space
>
> [3] OpenAI, 2024, O1 – https://openai.com/o1/
>
> [4] Guo et al., 2025, DeepSeek‑R1: Incentivising Reasoning Capability in LLMs via Reinforcement Learning

---

> > ### Comment · Reviewer_NeU4 · 2025-06-10
> >
> > I want to thank the authors for their new experiments and clarification. I'd like to keep my positive score.

---

### Official Review · Reviewer_eCiG · 2025-05-14

**Rating:** 6
**Confidence:** 4
**Ethics Flag:** 1

**Summary:**

The paper proposes a novel methodology to enable LLMs to reason/think in latent space instead of language tokens as is commonly done by CoT.
They purport the benefits of thinking in a latent space are that a model need not commit to a specific path ahead of time and can explore tree-like search over many paths at once. Secondly, it can also be more efficient to think in the latent space where efficiency is measured in terms of the number of forward passes of the Transformer required to achieve a certain quality.

The proposed method, COCONUT (Chain Of CONtinUous Thought), has two thinking modes - language space and latent space. In the latent space, instead of decoding the final embedding into an explicit token, we pass it as is to the next forward pass of the transformer.
To train COCONUT, the authors implement a multi-stage curriculum as follows. We start with a model trained on regular CoT data. We assume the CoT data splits into a number of reasoning steps. In each stage we replace the first k reasoning steps, with kc continuous thoughts (where c is a hyperparameter controlling the number of latent thoughts per reasoning step). Then the loss is optmized while masking the question and the continuous thoughts.
Special beginning and end marker tokens are used to demarcate between continuous thought and language thought modes.
The authors then proceed to show the efficacy of COCONUT first on a synthetic dataset ProsQA and then on a more general LLM reasoning task GSM8k.

**ProsQA analysis**:
The dataset requires the model to perform a graph search on a DAG to figure out the path to the answer. The authors take a pre-trained GPT-2 model and apply COCONUT fine-tuning on top of it specific to the ProsQA dataset. Compared to CoT where the model is trained to commit to a path right from the beginning, COCONUT allows the model to explore many different paths in a BFS-like manner and achieve a higher final accuracy of reaching the goal.
The authors also perform an interesting interpretability analysis to support that COCONUT's continuous thoughts are performing a tree-like search.

**General Reasoning Tasks**:
The authors focus on GSM8k and PRontoQA as the reasoning datasets for this part of the analysis. They compare with a CoT, No-CoT and iCot baselines. In addition, they ablate performing COCONUT without the stage-wise curriculum, and with a fixed pause token as the thought as well.
Overall, they find that on GSM8k, COCONUT significantly improves over No-CoT (but still underperforms CoT). However at a fixed token budget it is able to beat CoT since CoT requires a lot of intermediate tokens for its superior accuracy to show up.

**Questions To Authors:**

1. Could you elaborate more on the KV cache reuse you mention in line 153?

1. Could you also elaborate a bit on the different techniques you tried for changing optimizer state across stages?

1. Can this method work in the absence of CoT data?

1. Does this need to be done for each dataset separately? A more general recipe?

**Reasons To Accept:**

1. The idea is novel and the thoughtful and detailed experiments with the synthetic dataset show that the approach indeed has the ability to teach the model to explore multiple solution paths in parallel which is main drawback of CoT reasoning.

1. The paper studies an important and relevant problem. It attempts to address a focal drawback of CoT reasoning which can have a significant impact on the space of LLMs. Moreover, the paper is well-written and easy to read.

**Reasons To Reject:**

Overall, it feels like although the paper introduces a very nice technique, the experiments showing its efficacy are limited in scope.

1. The insightful analysis seems limited to only 2 datasets GSM8k and ProsQA out of which one is purely synthetic. ProntoQA seems very saturated at their model scale as even No-CoT gets over 90% accuracy.

1. Apriori it is unclear how the method can be applied to settings where CoT does not neatly decompose into reasoning steps.

1. It is also not clear whether the method can be applied in a general manner for all reasoning tasks or it needs dataset specific training for each dataset. Evidence for the possibility of the former could be built by evaluating on a wider suite of simple reasoning tasks.

---

> ### Author Response · Authors · 2025-06-02
>
> Thank you for the thoughtful review and for highlighting our method’s novelty, thorough experimentation, and clear writing. Below we address each of your questions.
>
> > General reasoning tasks
>
> Our main goal is to reveal and analyze the promising properties of latent reasoning, specifically its parallel search behaviour and superior compute efficiency. As discussed in the general response, we expect future works to scale up pretraining in latent space and improve the generality of the latent reasoning model.
>
> > Decompose into reasoning steps
>
> This is a great question and an interesting direction to explore. A natural first step is to segment the text into sentences. For example, Large Concept Models [1] apply Segment-any-Text (SaT) [2] for this purpose. A more flexible approach would be to dynamically decide thought boundaries during training, similar to the idea of Byte-Latent Transformer [3], which uses next-byte entropy to mark patch starts.
>
> > Dataset specific training / a more general recipe
>
> Currently, Coconut requires dataset-specific training. However, just as language space reasoning relied on task-specific training [4] before very strong LLMs came out, we view this as an intermediate stage. Our paper focuses on uncovering key properties of latent-space reasoning, and future latent-space pre-training might help eliminate the need for task-specific fine-tuning altogether.
>
> > KV cache reuse
>
> Consider a sequence of length $L$ containing $k$ continuous thoughts (indices $i$ … $i + k − 1$). Computing its loss requires $k + 1$ forward passes:
>
> - Pass 1 on tokens $[0, i)$ produces the hidden state at $i − 1$ (first continuous thought).
> - Pass 2 on $[0, i + 1)$ reuses the KV cache from pass 1 for $[0, i)$ and processes only the new position $i$.
> - Subsequent passes extend similarly, each reusing all prior KV entries.
>
> By reusing the KV cache from the previous forward pass, we could avoid repetitive computation.
>
> > Changing optimizer state
>
> Following Deng et al. [6], we reset AdamW’s moments to avoid stale momentum when gradients change direction at each stage switch. We did not explore more options in the optimizer setting.
>
> > In the absence of CoT data
>
> At the current stage, using CoT data to guide latent reasoning is still necessary. As shown in the ablation setting “Coconut w/o curriculum”, if we directly train the model with the final answer, without the guidance from CoT, the performance is significantly worse. We expect latent space pretraining to mitigate this dependency in future work.
>
> Thanks again for your valuable feedback!
>
> ## Reference
> [1] LCM team, 2024, “Large Concept Models: Language Modeling in a Sentence Representation Space”
>
> [2] Frohmann, 2024, “Segment Any Text: A Universal Approach for Robust, Efficient and Adaptable Sentence Segmentation”
>
> [3] Pagnoni, 2024, “Byte latent transformer: Patches scale better than tokens”
>
> [4] Ling et al., 2017, “Program Induction by Rationale Generation: Learning to Solve and Explain Algebraic Word Problems”
>
> [5] Kobbe et al., 2021, “Training Verifiers to Solve Math Word Problems”
>
> [6] Deng et al., 2024, “From explicit cot to implicit cot: Learning to internalize cot step by step”

---

> > ### Comment · Reviewer_eCiG · 2025-06-05
> > **Thanks for the author response**
> >
> > Thank you for your response to my questions and concerns. I would recommend adding these clarifications and discussions to the main paper. While I understand that this paper is a first foray into latent reasoning, I will maintain my score due to my concerns regarding the dataset specificity of the recipe and limited evaluation within this paper.

---

### Official Review · Reviewer_BZuh · 2025-05-17

**Rating:** 6
**Confidence:** 4
**Ethics Flag:** 1

**Summary:**

This paper introduces a novel reasoning paradigm for LLMs called Coconut (Chain of Continuous Thought), which departs from conventional language-based reasoning (e.g., CoT) by operating directly in the latent embedding space of the LLM rather than the language space. Instead of decoding a hidden state into a token, Coconut feeds the hidden state back into the model as the next input, enabling latent-space reasoning steps that are unconstrained by language syntax or semantics. The authors demonstrate that this mechanism supports breadth-first search-style planning, encodes multiple reasoning paths simultaneously, and achieves superior results to CoT on logical and planning-intensive tasks such as ProntoQA and GSM8K. Coconut is shown to improve both accuracy and efficiency, with fewer reasoning steps and better final outcomes.

Pros:
This paper draws inspiration from human brains. It extends the LLMs to have the capability to think as it desires. The basic idea seems to be very impactful and transformative.

Cons:
The current results are quite premature. The GSM results are much worse at this point.

**Reasons To Accept:**

This paper draws inspiration from human brains. It extends the LLMs to have the capability to think as it desires. The basic idea seems to be very impactful and transformative.

**Reasons To Reject:**

I think the current results have not fully unleashed its potential. However, I don't think it's a deal breaker. I think this paper provides very good insights.

---

> ### Author Response · Authors · 2025-06-02
>
> Thank you for the thoughtful review and for describing Coconut as impactful and transformative!
>
> Regarding GSM8K performance, please see our general response for additional discussion. We agree that fully unlocking the potential of latent reasoning will require further research, and we are actively pursuing larger-scale latent space pretraining to close the gap.

---

> > ### Comment · Reviewer_BZuh · 2025-06-06
> > **Thanks for the author response**
> >
> > Thank you for your response. I think the paper started a new direction and more studies need to be done to push the results further. I remain positive about this paper.

---

### Author Response · Authors · 2025-06-02
**General Response**

We thank all reviewers for their careful and constructive feedback. We are encouraged that reviewers view latent space reasoning as both novel and promising. Below we address the common questions:

## Performance on real-world applications

Our paper’s primary aim is to reveal and analyse the distinctive properties of latent reasoning—its parallel search behaviour and superior efficiency. On GSM8k, although Coconut does not yet surpass language CoT in terms of accuracy, we present two promising results:

- **More favourable accuracy-efficiency curve**. When we curb the generation budget by skipping the first k reasoning steps, accuracy for pure language reasoning deteriorates quickly. Replacing those k steps with k continuous thoughts keeps performance largely intact. (Fig. 7, upper)

- **Gains from additional continuous thoughts**. Accuracy rises steadily as we increase the number of latent thoughts, indicating that continuous thoughts add real reasoning capacity. (Fig. 7, lower)

We agree that stronger performance on open‑domain tasks will require further research. Language‑space reasoning currently enjoys an edge because it directly taps into knowledge acquired during next‑token pre‑training. We therefore view **latent space pre‑training** as a crucial next step. Recent community work underscores its feasibility:


Ref. | Approach | Model / Data Size |
| - | - | - |
| [1]  | Adaptive looping of a compute unit before predicting the next token | 3.5 B params, 800 B tokens |
| [2]  | Predicting the embedding of next sentences encoded by SONAR [4] | 1.6 B diffusion model, 1.3 T tokens |
|  [3] |  Predicting the next “concepts” behind human language, which are extracted with a sparse autoencoder | 1.3 B model, 200 B tokens|

These studies provide early evidence that large‑scale latent computation can be learned under token‑level or sentence‑level supervision. We believe it’s a promising direction to combine Coconut with such pre‑training regimes to build richer latent spaces and, ultimately, stronger reasoning performance on real-world tasks.

## Reference:
[1]  Geiping et al., 2025, "Scaling Up Test-Time Compute with Latent Reasoning"

[2]  LCM Team, 2024, "Large Concept Models: Language Modelling in a Sentence Representation Space"

[3] Tack et al., 2025, “LLM Pretraining with Continuous Concepts”

[4] Duquenne et al., 2023, “SONAR: Sentence-Level Multimodal and Language-Agnostic Representations”

---

### Decision · Program_Chairs · 2025-07-08

**Decision:**

Accept

**Comment:**

This paper introduces Coconut, a novel and impactful approach for LLMs to perform reasoning in the continuous latent space, rather than solely through language tokens. The paper has received unanimously positive reviews.

The method enables more efficient and parallel exploration of reasoning paths, akin to breadth-first search. The detailed analysis, particularly with the synthetic ProsQA dataset, provides compelling evidence for Coconut's unique capabilities in managing multiple reasoning trajectories.

While reviewers initially raised valid concerns regarding current empirical performance on real-world tasks like GSM8k (though showing significant improvement over No-CoT), the reliance on dataset-specific training, and the use of GPT-2 only, the authors provided thorough and convincing rebuttals in the discussion period. They effectively framed these limitations as inherent to a pioneering work in an emergent field, outlining clear paths for future research (e.g., large-scale latent space pre-training, applicability to larger models like Llama 3 with new results, and system-level optimizations for scalability). The authors' additional experiments and clarifications further strengthened the paper, addressing most open questions. The reviewers recommended to study continuous thoughts further to fully realize the potential of this method. Additionally, it is recommended to study more domains beyond math and logical reasoning.

As such, I recommend acceptance.